

# Camera-trapping estimates of the relative population density of *Sympetrum* dragonflies: application to multihabitat users in agricultural landscapes

Akira Yoshioka[1], Toshimasa Mitamura[2], Nobuhiro Matsuki[3], Akira Shimizu[4], Hirofumi Ouchi[1], Hiroyuki Oguma[5], Jaeick Jo[1], Keita Fukasawa[5], Nao Kumada[5], Shoma Jingu[6] and Ken Tabuchi[7]

[1] Fukushima Regional Collaborative Research Center, National Institute for Environmental Studies, Miharu, Tamura-gun, Fukushima, Japan
[2] Hama-dori Research Centre, Fukushima Agricultural Technology Centre, Soma, Fukushima, Japan
[3] Aizu Research Centre, Fukushima Agricultural Technology Centre, Aizubange, Fukushima, Japan
[4] Unaffiliated, Ami, Ibaraki, Japan
[5] Biodiversity Division, National Institute for Environmental Studies, Tsukuba, Ibaraki, Japan
[6] Forestry and Forest Products Research Institute, Forest Research and Management Organization, Tsukuba, Ibaraki, Japan
[7] Tohoku Agricultural Research Center, National Agriculture and Food Research Organization, Morioka, Iwate, Japan

Corresponding author
Akira Yoshioka,
yoshioka.akira@nies.go.jp

## ABSTRACT

Although camera trapping has been effectively used for wildlife monitoring, its application to multihabitat insects (*i.e.*, insects requiring terrestrial and aquatic ecosystems) is limited. Among such insects, perching dragonflies of the genus *Sympetrum* (darter dragonflies) are agroenvironmental indicators that substantially contribute to agricultural biodiversity. To examine whether custom-developed camera traps for perching dragonflies can be used to assess the relative population density of darter dragonflies, camera trapping, a line-transect survey of mature adult dragonflies, and a line-transect survey of exuviae were conducted for three years in rice paddy fields in Japan. The detection frequency of camera traps in autumn was significantly correlated with the density index of mature adults recorded during the transect surveys in the same season for both *Sympetrum infuscatum* and other darter species. In analyses of camera-detection frequency in autumn and exuviae in early summer, a significant correlation was observed between the camera-detection frequency of mature adults and the exuviae-density index in the following year for *S. infuscatum*; however, a similar correlation was not observed for other darter species. These results suggest that terrestrial camera trapping has the potential to be effective for monitoring the relative density of multihabitat users such as *S. infuscatum*, which shows frequent perching behavior and relatively short-distance dispersal.

## INTRODUCTION

Technologies for automated and efficient monitoring of biodiversity are becoming increasingly important (*Bush et al., 2017*). Camera trapping has drawn increasing attention as a relatively reliable, non-intensive, non-invasive, and inexpensive tool (*Bilodeau et al., 2022*; *Delisle et al., 2021*).

Although farming is typically assumed to harm wildlife inhabiting natural habitats, semi-natural agricultural landscapes with a long history of cultivation typically play an important role in biodiversity conservation (*Kadoya & Washitani, 2011*; *Queiroz et al., 2014*). Notably, multihabitat users, which require multiple habitats to complete their life cycles, are critical components of biodiversity in agricultural landscapes (*Batáry et al., 2020*; *Kadoya, Suda & Washitani, 2009*; *Kadoya & Washitani, 2011*). Multiple ecosystems maintained by natural processes and anthropogenic activities with spatial and temporal heterogeneities have benefited from such organisms (*Batáry et al., 2020*; *Kadoya & Washitani, 2011*; *Yoshioka et al., 2017*). Notably, wetland animals such as birds, frogs, dragonflies, and other aquatic insects, which use both terrestrial and aquatic habitats, are expected to be critical components of biodiversity in historical rice cultivation zones, where wet rice paddy systems provide an alternative to natural wet habitats (*Kadoya, Suda & Washitani, 2009*; *Kadoya & Washitani, 2011*; *Katayama et al., 2019*).

However, few studies have evaluated the use of camera trapping to monitor multihabitat users other than mammals, particularly for small and ectothermic organisms such as insects (*Delisle et al., 2021*), though some smart image-based classification systems have been developed for pests or flower-visiting insects (*e.g.*, *Bjerge, Mann & Høye, 2022*). In general, camera trapping can be more easily applied to terrestrial ecosystems than to aquatic ecosystems owing to optical and physical issues and the associated increases in cost (*Bilodeau et al., 2022*). If terrestrial camera traps can indicate the dynamics of multihabitat users not only at the terrestrial life stage but also at the stage depending on the aquatic environment, they can enhance the effectiveness of camera trapping. This will be helpful in biodiversity monitoring and conservation in agricultural landscapes.

Darter dragonflies (*Sympetrum* spp.) can be considered suitable candidates for exploring the potential of camera trapping. They are typical multihabitat users and are known as biodiversity indicators of rice-dominated landscapes in Japan (*Inoue & Tani, 2010*; *Sprague, 2003*; *AFFRC, NIAES and NIAS, 2012b*; *Tanaka, 2016*). In spring, larvae typically grow in aquatic habitats, including water-irrigated rice paddy fields. Immature adults emerge in early summer, leave paddies, and stay in cooler habitats, such as forests or mountainous areas, during the summer. They then return to autumn paddies and oviposit as mature adults (*Corbet, 1999*; *Sugimura et al., 1999*; *Watanabe, Susa & Taguchi, 2005*). Mature adults have red bodies and are a familiar autumn feature in Japan known as *akatombo*, meaning "red dragonfly" (*Inoue & Tani, 2010*). However, comparisons of the current and past numbers of observed individuals at some locations and questionnaire surveys of odonatologists suggest that the population has rapidly declined in some regions (*Jinguji & Uéda, 2015*; *Nakanishi et al., 2020a*; *Nakanishi et al., 2020b*; *Ueda & Jinguji*,

*2013*). Mature adults exhibit a characteristic perching behavior on natural or artificial rod-like materials. *Yoshioka et al. (2020)* developed a camera-trapping system to automatically detect individuals perching on sticks using inexpensive and energy-saving light sensors. Although *Yoshioka et al. (2020)* revealed that camera traps set for two days at a research institute could detect perching darter dragonflies with considerable accuracy, it remains unclear whether they can be used to quantitatively assess and monitor the population status of dragonflies in agricultural landscapes.

In this study, we explored the effectiveness of camera trapping as a tool for assessing and monitoring the population dynamics of darter dragonflies as indicators of multihabitat users in agricultural landscapes. Using camera traps developed for perching darter dragonflies (*Yoshioka et al., 2020*), two key questions were examined: (1) whether camera trapping is as effective as the classical survey method (*i.e.*, a line-transect survey) for assessing the relative population density of mature adult darter dragonflies, and (2) whether the relative population density of mature adult dragonflies, as determined by camera trapping, indicates the relative population density of exuviae of emerging adult darter dragonflies (determined by aquatic environments). If these are true, we can expect a greater contribution from camera trapping to the monitoring of multihabitat users such as darter dragonflies.

## MATERIALS & METHODS

### Study area

The study was conducted in a farmland area in eastern Fukushima Prefecture, northeastern Japan (Fig. 1). The area is characterized by typical agricultural landscapes, including rice paddy fields, secondary forests, and woodlands. The area was enclosed within the following four sets of coordinates: (37°47′N, 140°34′E), (37°47′N, 140°59′E), (37°30′N, 140°59′E), and (37°30′N, 140°34′E). The annual mean temperature and annual precipitation during 1991–2020 obtained from the Fukushima Local Meteorological Observatory were 13.4 °C and 1207.0 mm, respectively (*Japan Meteorological Agency, 2022*). The study area was a part of the area targeted by a three-year survey of biodiversity in paddy fields where rice farming was restarted after the lifting of the evacuation order following the Fukushima Daiichi Nuclear Power Plant accident in 2011 (*National Institute of Informatics, 2022*). The study area included six study plots (Harimichi, Imada, Kanaya, Mimigai, Sakata, and Sugaya), each of approximately 0.7–1.2 ha comprising one or a few rice paddy fields (Fig. S1). We were granted verbal permission for conducting the research on each of the farmer's private fields. The farmers were not represented by an organization, but were in a cooperative relationship with the Fukushima Agricultural Technology Centre. Rice farming in Kanaya, Mimigai, Sakata, and Sugaya was suspended during the evacuation and restarted between 2014 and 2018; however, rice farming continued without interruption in Harimichi and Imada. The camera-trapping and dragonfly adult surveys in the autumn of 2018 were not conducted in the Imada plot. Additionally, the target rice paddy fields in Kanaya were changed to neighboring rice paddy fields in 2019 (Fig. S1) because the farmers intended to change farming methods.
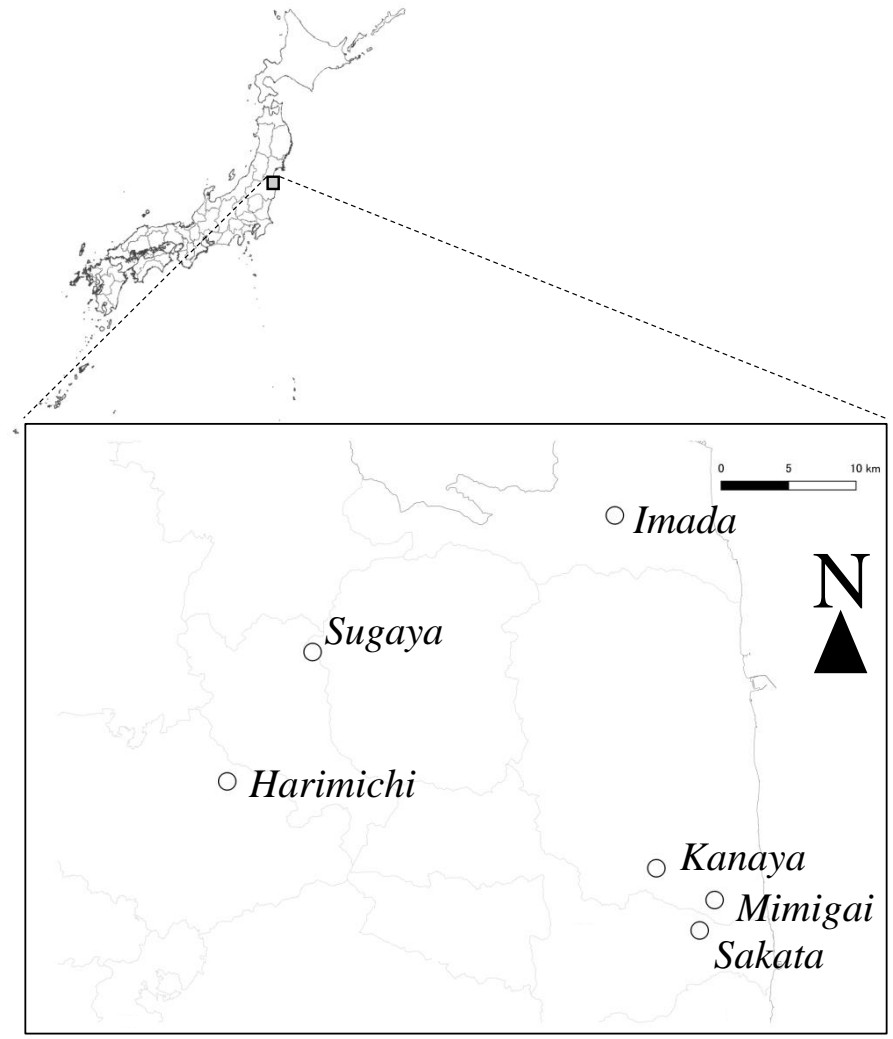

Source of the background maps:
Geospatial Information Authority of Japan

**Figure 1** **Study area.** Open circles show the study plots. The background maps were obtained from the *Geospatial Information Authority of Japan (2012)* following the Geospatial Information Authority of Japan Website Terms of Use (https://www.gsi.go.jp/ENGLISH/page_e30286.html, accessed 2022-11-11), which is compatible with the Creative Commons Attribution License 4.0 (https://creativecommons.org/licenses/by/4.0/legalcode, accessed 2022-11-11).

## Camera-trapping system for darter dragonfly detection

*Yoshioka et al. (2020)* developed the original camera trap system based on the following principles. A perching dragonfly changes the resistance value of an upper light sensor at the tip of a rod-shaped detector relative to a bottom light sensor, changes the difference in signals between the sensors over a certain period, and activates a connected camera to a certain extent, thereby triggering a picture to be taken of the tip of the detector. In addition, if an image is captured, an additional picture is not taken unless the change in the relative

signal values between sensors is restored because of dragonfly takeoff. In a previous study, a change of 5% and waiting period of 10 s (8 s for camera rebooting and an additional 2 s for image capture) could adequately detect perching darter dragonflies.

The camera trap used in this study comprised a detector section, processor section, and a connected commercial digital camera (TLC 200; Brinno, Inc., Taipei, Taiwan). Two CDS cells with a dark resistance of 1 M $\Omega$ were used as light sensors in the detector section. The upper cell was then covered with an LED silicone cap. The bottom of each cell was colored black using a permanent marker. These cells were connected to a microprocessor *via* USB cables. An acrylic pipe (outer diameter: 10 mm; inner diameter: seven mm; length: 500 mm) was used to enclose the cells. The top of the pipe was surrounded by a 150-mm-wide masking tape as a transmission diffuser and 20-mm-wide black vinyl tape and was capped with a 0.5-mm-thick polyethylene terephthalate plate using an adhesive.

In the processor section, an 8-bit PIC microcontroller (Microchip Technology Inc., Chandler, AZ, USA) was used as the main microprocessor. Signal values of 0–1,023 were obtained, which were processed to determine the presence or absence of a perching dragonfly based on the resistance values for each CDS cell in the detector section, which decreased with increasing illumination intensity. More details about the algorithm and materials used in the camera trap can be found in *Yoshioka et al. (2020)*.

During the study, the exterior of the processor section was modified to make it easier to check the status of the traps and (re)boot them. Thus, three versions of the camera trap with different exterior materials for the processor section were used (Fig. S2). Version 1 was characterized by a processor section enclosed in a gray polyvinyl chloride (PVC) pipe to prevent rainwater intrusion (Fig. S2A). Two silicone caps, penetrated by the detector section, were inserted at the top of the PVC pipe. The enclosed microprocessor and its switch were attached. Investigators could not see or touch the microprocessor without removing the PVC pipe, which made it difficult to handle the camera trap. Version 2 (Fig. S2B) included a transparent PVC pipe surrounding the microprocessor and microprocessor switch, which was separated from the microprocessor and enclosed in a PVC pipe with a larger diameter below the PVC pipe enclosing the microprocessor and connected to the microprocessor *via* cables. Investigators could visually check the LED light on the microprocessor (which blinked if it worked correctly). The switch in the separate larger-diameter pipe could be handled relatively easily. However, the part connecting the microprocessor to the switch was vulnerable to corrosion owing to moisture during long-term use. In version 3 (Fig. S2C), the microprocessor and its switch were enclosed in a translucent 400 mL polypropylene bottle for salad dressing (MT-TORIMATSU Inc.). The bottle was attached upside down to the PVC pipe with a detector. Cables connected the processor and detector parts through the bottle spout. Investigators could quickly check and handle the processor. Preliminary experiments revealed that the microprocessor was adequately waterproof. In addition, versions 1 and 2 used three AA nickel hydride batteries for the processor, whereas version 3 used three AAA nickel hydride batteries. In the survey, the PVC pipe with the detector part was attached to a tripod or woodpile to adjust the height of the camera trap (approximately 2 m), regardless of the camera trap version.
## Darter dragonfly survey using camera traps

A camera-trapping survey was conducted from 2018 to 2020 in the study plots. For each year in each plot, three camera traps were set up from the end of August to early September, and data were collected from the end of October to early November (Table S1). To cover a large area within a plot without disturbing farming, three camera traps were located on the levees of the paddy fields so that they were as far from each other as possible within the plot (Fig. S1). Although the traps were removed at the end of the study season every year, the levees where trap were set were consistent across years, and each trap location in each plot was assigned a location ID. When the trap location was changed at the field level, as in the case of Kanaya, a new location ID was assigned. Different versions of the camera trap were equally allocated to each plot, to the maximum extent possible. Every year, the detector parts of each version were randomly allocated to plots and locations to minimize the effects of individual camera traps. Each camera trap was inspected during early October. We manually intercepted the top of the detector and checked it for successful autodetection. In addition, after the transect surveys or typhoon events, the camera traps were briefly checked to ensure that they did not fall or run out of battery power. When necessary, the traps were reset by rebooting camera traps, exchanging batteries, and exchanging camera traps.

All pictures automatically captured by the camera traps and recorded in SD memory cards in the cameras were visually checked, and those correctly capturing darter dragonflies were recorded as darter dragonfly detection events (hereinafter, the frequency of these events is called "camera-detection frequency"). If two pictures were captured within 5 s, the former was removed from the analyses to avoid overestimating dragonfly detection following Yoshioka et al. (2020). Because the signal to wake up the camera was sent a few seconds before the signal to capture a picture, an extra picture was obtained if the camera was already active.

*S. infuscatum* (noshime-tombo, in Japanese) was recorded based on its characteristic dark spot on the tip of each wing; this species can be easily distinguished from other darter species (Yoshioka et al., 2020). Because the identification of other darter dragonflies at the species level was difficult owing to low image resolution, they were pooled in subsequent analyses. Most other darter dragonflies were likely the autumn darter *S. frequens* because this species was dominant based on a transect survey, and we did not directly observe darter dragonflies other than *S. frequens* and *S. infuscatum* perching on the camera traps during the survey season. These two species are typical darter dragonflies reproducing at rice paddy fields in Japan (Jinguji, Ozaki & Uéda, 2019) and are considered suitable as biodiversity indicator species of paddy fields (AFFRC, NIAES and NIAS, 2012b). Both males and females of these species are known to exhibit perching behavior (Miyakawa, 1994; Watanabe, Matsuoka & Taguchi, 2004), while males do not clearly exhibit territorial behavior (Taguchi & Watanabe, 1986; Sugimura et al., 1999; Watanabe, Matsuoka & Taguchi, 2004). In addition, mark-recapture studies have shown that the recapture rates of these species are generally quite low (Taguchi & Watanabe, 1986; Watanabe, Matsuoka & Taguchi, 2004; Fukui, 2012) and it is considered unlikely for a particular individual to stay perched at the same point for more than a day (Fukui, 2012).

Owing to external factors, such as adverse weather and unanticipated internal factors, the period for which the camera traps worked effectively varied. To standardize the dragonfly detection frequency among camera traps, the occasion index (*i.e.*, the period during which each camera trap effectively worked each year) was also required. To calculate this value, the dates on which pictures were taken were obtained; history of camera trap setting, resetting, and collection were checked; and periods starting with the camera trap (re)setting and ending with the last picture before resetting (or successfully capturing a picture through an operation check at the end of the survey) were extracted. "Last pictures" included pictures with an appropriate view of the tip of the detector, regardless of the presence of a dragonfly (pictures captured from a fallen camera were invalid because such traps were considered ineffective). Although the exact date when camera traps became ineffective could not be determined, the date of the last picture before resetting is typically used in camera trap monitoring (*Fukasawa et al., 2016*). In this study, autodetection (irrespective of whether the dragonflies were correctly detected) typically occurred daily when the camera trap was active (*Yoshioka et al., 2020*). Second, the length of each period was calculated as the difference between the date of (re)setting the camera trap and the last date when a picture was captured before reset or the date of camera trap collection. Finally, the length of each segment was summed as the number of occasions per camera trap per year. Thus, the occasion index at the location in the year can be formulated as follows:

$$occasion = \sum_{i=1} \left( d_i^{last} - d_i^{set} \right)$$

where $d_i^{last}$ is the date of the last picture in the $i$-th segment and $d_i^{set}$ is the date of the (re)setup of the camera trap.

For example, consider the case in which a camera trap was set up on September 5, reset in response to a technical issue identified on September 19, and collected on October 31 with depleted batteries. The last picture before September 19 was captured on September 11, and the last picture before October 31 was captured on October 30. In this case, the occasion index for the trap was (September 11 − September 5) + (October 30 − September 19) = 6 + 41 = 47. If a camera trap remained active without any issues, the number of days between the date of setup and the date of collection corresponded to the occasion index.

## Transect survey of adult darter dragonflies in autumn

A line-transect survey was conducted as a classical method for measuring the relative population density of adult dragonflies (*Bouwman et al., 2009*; *Pearce-Higgins & Chandler, 2020*). This survey was conducted in the autumn of 2018, 2019, and 2020. Each study plot was surveyed twice (once each in September and October) from 9:30 to 17:00 h on days with mild wind and no measurable precipitation. According to the number of paddy fields in the plots, 8–16 line transects of 10 × 2 m were set along the levees of the rice paddy fields in each plot. Four line transects per rice field were set, except for the Mimigai plot, which had one large rice paddy field. During the line-transect survey, the investigator slowly (about 0.15 m/s) walked in one direction while searching for dragonflies within the line transect, and the number and species of darter dragonflies were recorded for each transect. When

relatively small and inconspicuous individuals, such as females of *S. kunckeli* (maiko-akane in Japanese) or *S. eroticum* (foot-tipped darter), were observed, a sweeping net of 50 cm in diameter was used to capture and evaluate them. For each year and plot, the adult-density index by line transects for *S. infuscatum* was obtained for analysis as follows:

$$\text{adult-density index by line transects} = \left( \frac{n_{\text{inf1}}}{n_t} + \frac{n_{\text{inf2}}}{n_t} \right) / 2,$$

where $n_{inf1}$ and $n_{inf2}$ are the total numbers of observed *S. infuscatum* in the first and second surveys, respectively, and $n_t$ corresponds to the total number of line transects in the plot per survey. In the same way, the adult-density index by line transects of other darter dragonflies (*i.e.,* darter dragonflies other than *S. infuscatum*) was calculated.

## Transect survey of exuviae of darter dragonflies in early summer

Population growth of darter dragonflies may be related to the quality of the local aquatic environment rather than to the abundance of adults with high mobility that oviposit at poor-quality aquatic sites (*Raebel et al., 2010*). Therefore, sampling exuviae of emerged adults is a highly effective method for surveying viable populations of dragonflies and the quality of local aquatic environments, although it has some disadvantages, such as low detectability (*Pearce-Higgins & Chandler, 2020*). This method has been recommended for surveys of darter dragonfly densities in rice paddy fields in Japan (*Baba, Kusumoto & Tanaka, 2019*; *Mitamura et al., 2013*; *Nakanishi et al., 2022*). In this study, Mitamura et al. (T. Mitamura, N. Matsuki, A. Yoshioka, K. Tabuchi, 2020, unpublished data) collected a part of the dataset to survey the biodiversity in rice paddies after the Fukushima nuclear disaster. The data set for darter exuviae was obtained following a sampling protocol based on a survey and evaluation manual for indicator animals of functional agrobiodiversity compiled by the national government of Japan (*AFFRC, NIAES and NIAS, 2012a*; *Mitamura et al., 2013*). In this survey, four 10 m line transects per rice field were set along the levees of the rice paddy fields in each plot. The investigators visited each plot twice a year during the daytime, from late June to early July in 2018–2020. During the line-transect survey, an investigator walked slowly in one direction while searching for and collecting exuviae of dragonflies on rice plants up to the third row of rice hills from the edge of the levees (corresponding to a width of approximately 1 m). The collected exuviae were identified to the species level in the laboratory. The number of exuviae in the two surveys was summed for each transect in each year (hereinafter, "exuviae-density index by line transects"). This index was expected to show the relative density of the exuviae rather than the absolute density.

## Statistical analyses

To examine whether the camera-detection frequency of adult darter dragonflies reflected the adult-density index by line transects, a generalized linear mixed model (GLMM) with a negative binomial error distribution was constructed. In the GLMM, the camera-detection frequency of *S. infuscatum* ($n = 51$) was the response variable, adult-density index by line transects of *S. infuscatum* was the explanatory variable, and the log of the occasion index was the offset term. The year, study plot, location ID of a trap nested within a

study plot, and trap version were included as random effects. The adult-density index by line transects of *S. infuscatum*, was log-transformed before the GLMM to obtain a more likely relationship between the density of dragonflies and the camera-detection frequency as suggested by *Yoshioka et al. (2020)*. That is, a larger abundance of dragonflies would not always correspond to a higher camera-detection frequency when the number of camera traps is smaller than the abundance of dragonflies because only a small portion of individuals can occupy the detector in a short term. Therefore, log transform to reduce the weight of larger values of the adult-density index by line transects was desirable to obtain a more proportionate relationship with camera-detection frequency. Because the data of *S. infuscatum* included zero values, 0.5 was added before log transformation (*i.e.,* log(adult-density index by line transects of *S. infuscatum* + 0.5)) following the recommendation of *Yamamura (1999)*. Similarly, the effect of the adult-density index of other darter dragonflies on their camera-detection frequency was examined.

In addition, relationships between exuviae surveyed in early summer, as an indicator of the quality of the local aquatic environment, and adult darter dragonflies detected by camera traps in the autumn were examined. The exuviae-density index by line transects of *S. infuscatum* for each line transect in each year was divided by two to obtain the mean of the two surveys and averaged to obtain the mean exuviae density of *S. infuscatum* (the explanatory variable). The effect of the mean density of *S. infuscatum* exuviae on the camera-detection frequency of *S. infuscatum* in the same year was examined using a GLMM with a negative binomial error distribution ($n = 51$). The log-transformed occasion index was included as the offset term, and the year and location ID of a trap nested within the study plot were used as random effects. The effect of the camera-detection frequency of *S. infuscatum* on the exuviae-density index of the transects of *S. infuscatum* in the following year was also examined. The camera-detection frequency divided by occasion index was averaged in each plot for each year as the mean detection frequency per occasion. A GLMM with a negative binomial error distribution was then constructed using the exuviae-density index by line transects of *S. infuscatum* for each transect in each year as a response variable (the sum of values from two surveys was used rather than the mean because integer values were required for the GLMM with negative binomial) and the mean detection frequency per occasion of the previous year as an explanatory variable ($n = 116$). In the model, year and transect nested within the rice paddy fields and study plots were included as random effects. Similarly, the relationships between the exuviae-density index of other darter dragonflies and detection frequencies of these species were examined. The GLMM was implemented using the glmmTMB function in the glmmTMB package version 1.1.2 for R version 4.0.3 (*Brooks et al., 2017*; *R Core Team, 2020*). See Code S1 for details of the GLMM.

## RESULTS

Our camera traps detected 23,084 pictures of darter dragonflies during autumn over three years (Table S2). The most detected dragonflies were "other darter dragonflies" (*i.e.,* species other than *S. infuscatum*). The occasion index for camera traps (mean ± standard

**Table 1 Results of GLMM which estimated the effects of log-transformed mean adult-density index by line-transect survey on camera-detection frequency for (a) S. infuscatum and (b) other darter dragonflies.**

| Explanatory variables (Fixed effects) | Estimated value | Standard Error | Z value | P value (Z) |
|---|---|---|---|---|
| (a) | | | | |
| Intercept | 2.45 | 1.43 | 1.72 | 0.0862 |
| Mean adult-density index of S. infuscatum (log-transformed + 0.5) | 5.13 | 1.38 | 3.71 | 0.000209 |
| (b) | | | | |
| Intercept | 0.508 | 0.379 | 1.34 | 0.180 |
| Mean adult-density index of the other darter *dragonflies* (log-transformed + 0.5) | 1.68 | 0.566 | 2.97 | 0.00296 |

deviation) was 42.76 ± 11.14 days. From the line-transect surveys of adults in the autumn, the summer darter *S. darwinianum*, banded darter *S. pedemontanum*, *S. kunckeli*, and *S. eroticum* were recorded in addition to the major species *S. infuscatum* and *S. frequens* (Table S3). *S. frequens* was the most abundant species in most plots, whereas *S. infuscatum* was the most abundant species in the Harimichi plot. In line-transect surveys in early summer, exuviae of *S. infuscatum* were documented in only Harimichi and Imada plots (Table S4). Most of the sampled exuviae were *S. frequens*; however, *S. infuscatum*, *S. darwinianum* and *S. kunckeli* were also obtained. All *Sympetrum* species observed in this study were univoltine (*Inoue & Tani, 2010*). Therefore, the exuviae in the following year corresponded to the next generation of the adults detected in the autumn.

A GLMM revealed significant positive relationships between the camera-detection frequency and adult-density index by line transects ($p = 0.000209$ and $0.00296$, respectively) for both *S. infuscatum* and other darter dragonflies (Table 1; Fig. 2; Table S5). The estimated random effects (±conditional standard deviation) for each level of the trap version by the *ranef* function (Code S1) were very small (*S. infuscatum*: version 1 $-2.00 \times 10^{-8} \pm 2.08 \times 10^{-4}$, version 2 $1.77 \times 10^{-8} \pm 2.01 \times 10^{-4}$, version 3 $-1.41 \times 10^{-9} \pm 1.75 \times 10^{-4}$; other darter dragonflies: $3.91 \times 10^{-12} \pm 3.67 \times 10^{-6}$, $-4.64 \times 10^{-12} \pm 4.29 \times 10^{-6}$, $2.86 \times 10^{-13} \pm 1.28 \times 10^{-6}$, respectively) relative to the fixed effects in Table 1. The relationships between exuviae and camera-detection frequencies in the same year were not significant for *S. infuscatum* or for other darter dragonflies (Table 2; Fig. 3). A significant positive relationship between the camera-detection frequency and the exuviae-density index in the following year was obtained for only *S. infuscatum* (Table 3; Fig. 4). The analyses using the adult-density index and the exuviae-density index of *S. frequens* instead of those of other darters showed similar results (Table S8).

## DISCUSSION

Our results demonstrated that the detection frequency of darter dragonfly adults in autumn using camera traps was positively correlated with the density index of adults obtained by transect surveys for both *S. infuscatum* and other darters. Thus, camera trap data represent a quantitative indicator of the relative population density of mature adults of these dragonflies. A short-term (two days) analysis of camera traps (*Yoshioka et al., 2020*)

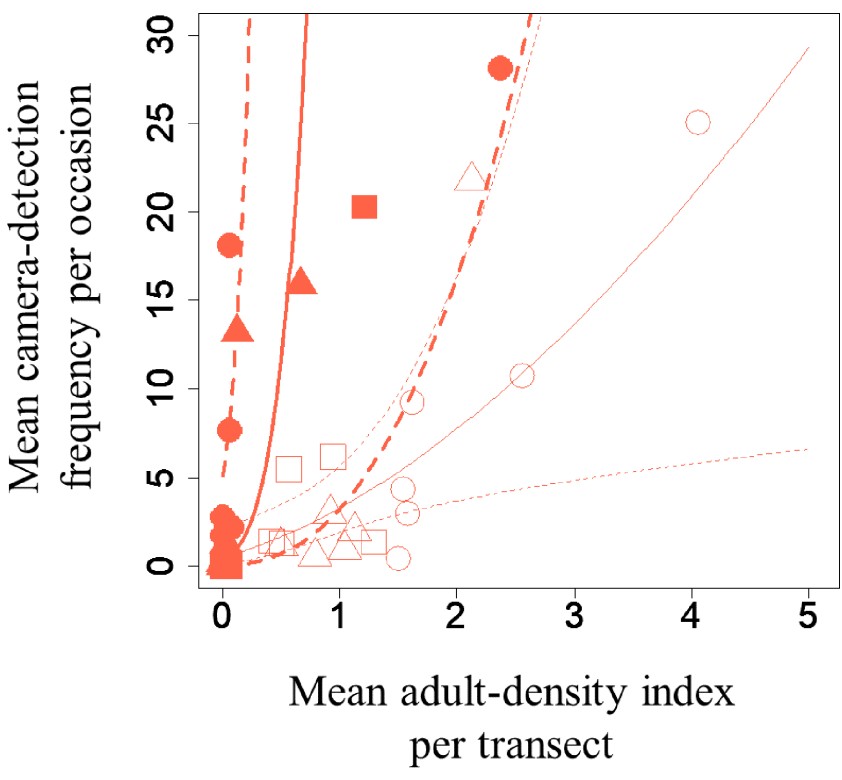

**Figure 2** **Relationships between the mean adult-density by line-transects of darter dragonflies and mean camera-detection frequency per occasion of them for each year each plot.** Solid and open symbols correspond to *S. infuscatum* and other darter dragonflies, respectively. Squares, circles, and triangles correspond to 2018, 2019, and 2020, respectively. When a statistically significant effect of the explanatory variable was obtained, the prediction curve (solid line) and 95% confidential intervals (dashed lines) by GLMM (all random effects were set to zero, and new random effect levels were not considered for simplicity) were also plotted. Thick and thin lines correspond to predicted values for *S. infuscatum* and other darter dragonflies, respectively.

**Table 2** **Results of GLMM which estimated effects of mean exuviae density index by line-transects in early summer on camera-detection frequency for adults of (a) *S. infuscatum* and (b) other darter dragonflies in autumn of the same year.**

| Explanatory variables (Fixed effects) | Estimated value | Standard Error | Z value | *P* value (Z) |
|---|---|---|---|---|
| (a) | | | | |
| Intercept | 0.447 | 1.46 | 0.307 | 0.759 |
| Mean exuviae-density index of *S. infuscatum* | −0.456 | 2.63 | −0.173 | 0.862 |
| (b) | | | | |
| Intercept | 1.36 | 0.488 | 2.789 | 0.00529 |
| Mean exuviae-density index of the other darter dragonflies | 0.00820 | 0.0234 | 0.350 | 0.726 |

suggested that the correlation between the camera-detection frequency and observational survey results is weak. Thus, the camera-detection frequency might not reflect the true density of dragonflies, owing to the dominant perching behavior of a small number of individuals (*i.e.,* the detection frequency might be saturated against density). However,

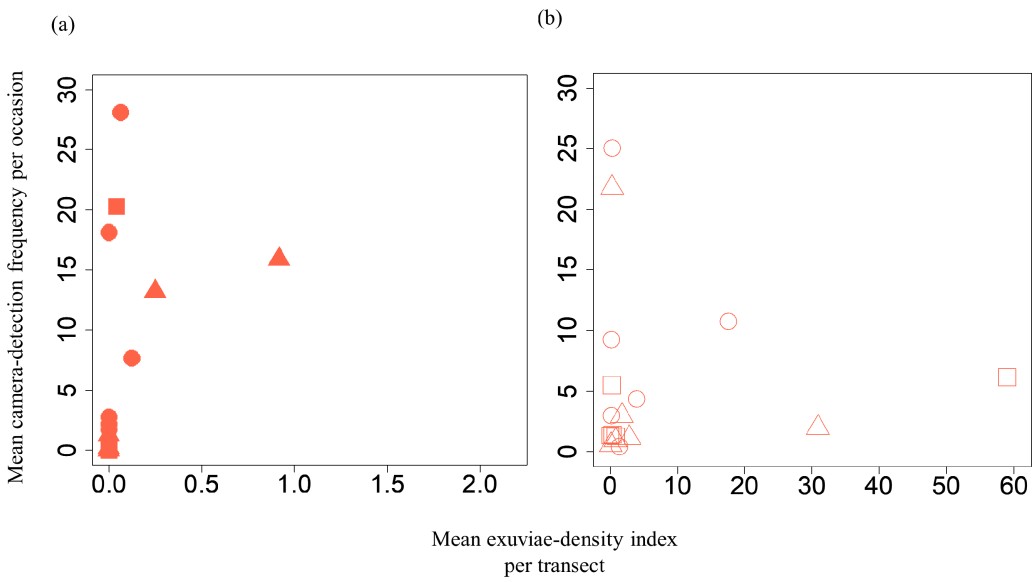

**Figure 3** Relationships between the mean exuviae-density index per transect obtained from the line-transect survey in early summer and mean camera-detection frequency per occasion in the same year. (A) *S. infuscatum* and (B) other darter dragonflies. Squares, circles, and triangles correspond to 2018, 2019, and 2020, respectively.

**Table 3** Results of GLMM which estimated the effects of mean camera-detection frequency of adult individuals in autumn of the previous year on mean exuviae-density index in early summer for (a) *S. infuscatum* and (b) other darter dragonflies.

| Explanatory variables (Fixed effects) | Estimated value | Standard Error | Z value | *P* value (Z) |
|---|---|---|---|---|
| (a) | | | | |
| Intercept | −8.79 | 3.04 | −2.89 | 0.00384 |
| Mean camera-detection frequency per occasion of *S. infuscatum* in the previous year | 0.333 | 0.110 | 3.02 | 0.00254 |
| (b) | | | | |
| Intercept | 0.926 | 0.673 | 1.38 | 0.169 |
| Mean camera-detection frequency per occasion of the other darter dragonflies in the previous year | −0.0228 | 0.0264 | −0.86 | 0.388 |

over a longer study period (exceeding one month), our findings suggest that the detection frequency obtained using adequate traps (three traps per plot in our study) can accurately reflect the log-transformed density index of adult dragonflies measured by line-transect surveys. The results for other darters here were considered to mainly reflect the density of *S. frequens*, given their large abundance in the line-transect surveys, field observations of perching behavior on the traps, and the results of analyses using data of *S. frequens* in the line-transect surveys instead of other darters. Therefore, the results suggest applicability of our camera trap to the mature adults of *S. infuscatum* and *S. frequens* during the autumn in our study system. These two species are known to be typical and symbolic biodiversity indicator species of rice paddy fields in Japan (*AFFRC, NIAES and NIAS, 2012b*; *Jinguji, Ozaki & Uéda, 2019*). The camera trap version, which changed only the processor section

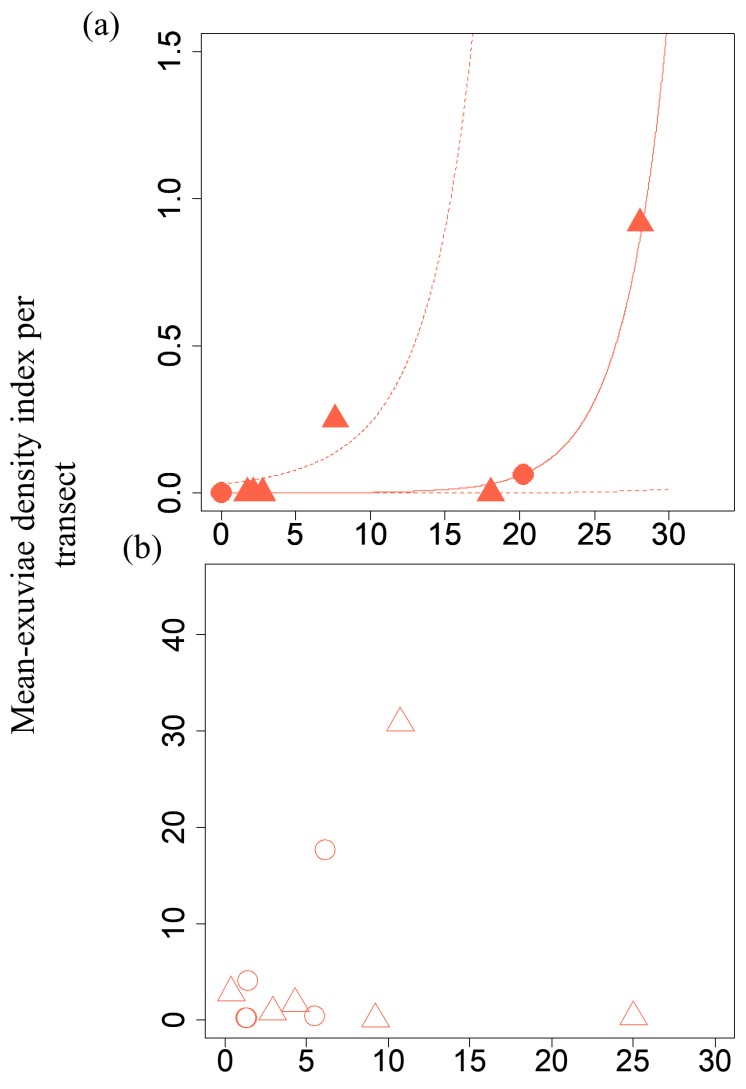

Mean-exuviae density index per transect

Mean camera-detection frequency per occasion of the previous year

**Figure 4 Relationships between the mean exuviae-density index per transect obtained by line-transect survey in early summer and mean camera-detection frequency per occasion in the autumn of the previous year.** (A) *S. infuscatum* and (B) other darter dragonflies. Circles and triangles correspond to 2019 and 2020, respectively. Because the result of GLMM for *S. infusactum* exhibited a statistically significant effect of the explanatory variable, the prediction curve (solid line) and 95% confidential intervals (dashed lines) were plotted (all random effects were set to zero, and new random effect levels were not considered for simplicity).

and did not change the detector part of the trap, was suggested to have hardly any effect on the results, as shown by the small random effects relative to the fixed effects.

Although the detection frequency of camera traps reflected the population density of mature adults, it was not predicted by the density of exuviae of new darter adult dragonflies. These results suggest that the density of exuviae is not clearly related to the density of adult

*S. infuscatum* or other species in autumn of the same year. Adult darter dragonflies may be substantially affected by environments other than farmlands (*e.g.*, woodlands) that are inhabited by individuals at the immature stage until they migrate to farmlands. However, our study was limited to farmlands and did not evaluate different habitats.

Furthermore, the detection frequency of *S. infuscatum* was positively related to the exuviae density in the following year; however, a similar relationship was not observed for other darter dragonflies. These species-dependent relationships may be explained by differences in dispersal ecology between *S. infuscatum* and *S. frequens* (the latter was the dominant species among other darter dragonflies in our study). Specifically, *S. frequens,* which migrates to farmlands in autumn from distant cooler areas several dozens of kilometers away (*Corbet, 1999*), makes it possible to maintain a high density of mature adults in "sink" farmlands with a less suitable aquatic environment for larvae. However, *S. infuscatum*, which returns to nearby paddy fields from surrounding forests (*Watanabe, Matsuoka & Taguchi, 2004*; *Watanabe, Susa & Taguchi, 2005*), may not be able to maintain high population densities in farmlands with unsuitable aquatic habitats. Alternatively, considering that males of *S. frequens* are known to be more abundant than females during late autumn in paddy fields (*Taguchi & Watanabe, 1986*), if the number of females of *S. frequens* was unexpectedly small in a plot with apparently high abundance of adults, it might result in limited exuviae abundance of the next generation in the plot. Because we did not obtain adequate data on sex in this study, we could not exclude this hypothesis. However, *S. infuscatum* similarly shows higher male abundance in the paddy fields (*Watanabe, Matsuoka & Taguchi, 2004*). In addition, a previous study of *S. frequens* (*Taguchi & Watanabe, 1986*) showed that it is not rare that female abundance in autumn does not explain the abundance of new emergent adults at the same locations in the next year. Another likely and not mutually exclusive explanation is that low detectability in the exuviae survey (*Pearce-Higgins & Chandler, 2020*) makes detecting the relationships for other darter species difficult. *Mitamura et al. (2013)* reported that the exuviae density of darter dragonflies (mainly *S. frequens*) substantially varied within a few weeks. In our study, a small number of line transects showed extremely high densities, and many transects showed zero values. At the very least, however, the lower frequency of exuviae of *S. infuscatum* (found at only two plots during three years) compared with *S. frequens* appears to be due to low abundance in our study system and not due to lower detectability in the line-transect surveys than that of *S. frequens*, given that the size of their exuviae is similar (*Inoue & Tani, 2010*) and the timing of emergence of exuviae of these species is known to overlap (*Jinguji, Ozaki & Uéda, 2019*). Although the perching behavior of darter dragonflies in farmlands in early summer is considerably less conspicuous, camera trapping during this season may resolve this issue. Notably, our preliminary camera traps set in a few farmlands in early summer detected some darter dragonflies, although the frequency was lower than in autumn. In addition, discerning the exuviae of *S. infuscatum* from those of the summer darter *S. darwinianum* is relatively difficult owing to their morphological similarity (*Ozono, Kawashima & Futahashi, 2019*). Accordingly, camera trapping for adults is a complementary tool because adult *S. infuscatum* can be easily identified.

Although the results demonstrated that our camera trapping focusing perching behavior would be effective for surveying the relative population density of a few indicator species (*S. infuscatum*, and partly, *S. frequens*) in some paddy-dominated landscapes in Japan, there are many steps required for camera trapping to be alternative to, or supporting tool for, monitoring programs targeting whole communities of dragonflies. First, applicability of the traps to darters other than *S. infuscatum* and *S. frequens* remains to be demonstrated, though these two species are typical darters that reproduce in the paddy fields in Japan. Thus, application of the camera traps to areas with a higher density of other darter species requires a caveat. Males of some species may show more conspicuous territorial behavior and bias the camera-detection frequency. Also, improvement of hardware may be required. For example, it is necessary to set the camera closer to the top of detector to capture keys for identifying species and/or sex, which are usually found on the side of the thorax and the base or tip of the abdomen (*Sugimura et al., 1999*). In addition, there is a room to improve the effort required for application of our method. In this study, field work (setting, checking, and collecting traps) for camera trapping roughly required two person-hours for each plot (three camera traps) per season (autumn). However, the time required for visual inspection of all captured images was roughly estimated as 10 person-hours for each plot (three camera traps) per season. This appeared not to be time-efficient considering that line-transect surveys of adult darters took about one hour for each plot per season. Further, a long-term monitoring survey of the Odonata community by *Dolný, Pyszko & Šigutová (2021)*, for example, took eight (two hours × four visits) person-hours for each site per season (May to September). To conduct monitoring sustainably with limited effort, techniques for thinning pictures or automatic image classification will be required. Furthermore, percher species does not always prefer to stay on the top of a perch. Damselflies often stay on the side of a perch. In ecosystems with a more complex vegetation structure, the perch-like detector may be overlooked by dragonflies. To automatically capture them, development of other detectors or post-process of time-lapse pictures by automatic image classification technique may be alternatives. Not only the type of ecosystem, but also social aspects may also be important for the applicability of camera trapping. In this study, no camera trap was lost, but risk of vandalism may be not low in other countries or regions. Under such circumstances, the understanding and cooperation of local community members may be more important for camera trapping studies (*Sharma et al., 2020*).

Nevertheless, our camera trapping showed some merits compared with traditional field survey methods. Continuous monitoring would lead to high detectability. Our camera traps detected *S. infuscatum* at all the six plots, but line-transect surveys for adults and exuviae detected the species at five and two plots, respectively. In addition, camera trapping was also less affected by weather and the schedule of the trained investigator. Furthermore, evidence of monitoring could be stored in a digital format. Applicability to difficult-to-enter zones (such as remote islands or evacuation zones) will be a merit of our method; *Yoshioka et al. (2020)*, though this may be beyond the scope of our present study in the agricultural landscapes.

## CONCLUSIONS

Our results suggest that camera traps are effective not only for measuring the density index of adult darter dragonflies in terrestrial habitats but also, to some extent, for indicating exuviae density, which reflects aquatic habitat status. These results suggest the potential of applying camera traps to other multihabitat users inhabiting agricultural landscapes, such as waterfowls, amphibians, and other aquatic insects. Compared with terrestrial environments, camera trapping has been applied in aquatic environments to a lesser extent (*Bilodeau et al., 2022*, but see *McCleery et al., 2014*).

The applicability of terrestrial camera trapping to analyze population dynamics, including life stages in aquatic habitats, may depend on species and life history traits. Our study suggested that dispersal traits or spatial scale may be key factors explaining difference between *S. infuscatum* and other darters (mainly *S. frequens*). However, true population density and dynamics were not analyzed in this study and may require a mark-recapture technique. Although the recapture rate of the darters in Japan is generally low (*Taguchi & Watanabe, 1986*; *Watanabe, Matsuoka & Taguchi, 2004*; *Fukui, 2012*), our camera trap can serve as a good continuous observer because marks on the wings of perching dragonflies are expected to be relatively easily captured in a picture.

At present, our study has demonstrated the applicability of camera trapping to only a small number of biodiversity indicator species in limited landscapes. There is still room for improvement in automatic monitoring of Odonata diversity. Nevertheless, by revealing the benefits and limitations of camera trapping, our study provides a basis for developing a combination of efficient and sustainable biodiversity monitoring tools. For example, eDNA is a promising tool for efficiently identifying aquatic fauna; however, this approach is less effective for the quantification of population dynamics and community composition, owing to detection uncertainty (*Fukaya et al., 2022*). Combining screening of indicator species using eDNA metabarcoding and continuous and quantitative monitoring by camera traps customized to the species may be effective. In addition, underwater camera trapping, a developing technique for monitoring ocean biodiversity (*Bilodeau et al., 2022*), may be combined with the biodiversity monitoring of agricultural landscapes, including rice paddies. Surveys based on expert observations are expected to play an essential role in validating new tools. Complex combinations of monitoring tools will generate rich biodiversity data and robust statistical inferences (*Iijima, Nagaike & Honda, 2013*), providing a basis for evidence-based conservation policy-making.

## ACKNOWLEDGEMENTS

We are grateful to the farmers for their cooperation with the survey. We would like to thank Ms. W. Endo (Fukushima Agricultural Technology Center) for supporting our survey. We also appreciate the valuable and encouraging comments from the two reviewers (one anonymous and Prof. Adolfo Cordero Rivera).

### Funding

This work was supported by the Japan Society for the Promotion of Science (JSPS) KAKENHI grants nos. 18K05931 and 21H03656. The funders had no role in study design, data collection and analysis, decision to publish, or preparation of the manuscript.

### Grant Disclosures

The following grant information was disclosed by the authors:
Japan Society for the Promotion of Science (JSPS) KAKENHI: 18K05931, 21H03656.

### Competing Interests

Our institution (National Institute for Environmental Studies, Japan) has obtained a national patent (JP 6558701, Available at https://www.nies.go.jp/kenkyu/patent/jqjm1000000lcbm9-att/171_JPB_006558701.pdf) for the principal idea of perching dragonfly detection using passive light sensors.

### Author Contributions

- Akira Yoshioka conceived and designed the experiments, performed the experiments, analyzed the data, prepared figures and/or tables, authored or reviewed drafts of the article, and approved the final draft.
- Toshimasa Mitamura conceived and designed the experiments, performed the experiments, analyzed the data, authored or reviewed drafts of the article, and approved the final draft.
- Nobuhiro Matsuki conceived and designed the experiments, performed the experiments, analyzed the data, authored or reviewed drafts of the article, and approved the final draft.
- Akira Shimizu conceived and designed the experiments, authored or reviewed drafts of the article, and approved the final draft.
- Hirofumi Ouchi performed the experiments, analyzed the data, prepared figures and/or tables, and approved the final draft.
- Hiroyuki Oguma conceived and designed the experiments, authored or reviewed drafts of the article, and approved the final draft.
- Jaeick Jo performed the experiments, authored or reviewed drafts of the article, and approved the final draft.
- Keita Fukasawa conceived and designed the experiments, analyzed the data, prepared figures and/or tables, authored or reviewed drafts of the article, and approved the final draft.
- Nao Kumada conceived and designed the experiments, authored or reviewed drafts of the article, and approved the final draft.
- Shoma Jingu analyzed the data, prepared figures and/or tables, and approved the final draft.
- Ken Tabuchi conceived and designed the experiments, performed the experiments, prepared figures and/or tables, authored or reviewed drafts of the article, and approved the final draft.

## Field Study Permissions

The following information was supplied relating to field study approvals (*i.e.*, approving body and any reference numbers):

We were granted verbal permission for conducting the research on each of the farmer's private fields.

The farmers were not represented by an organization, but were in a cooperative relationship with the Fukushima Agricultural Technology Centre.

We have no form of written permission for the conduction of the research.

## Patent Disclosures

The following patent dependencies were disclosed by the authors:

FLYING ORGANISM DETECTION DEVICE/JP 6558701/26th July 2019: Available at https://www.nies.go.jp/kenkyu/patent/jqjm1000000lcbm9-att/171_JPB_006558701.pdf.

## Data Availability

Code and raw data are available in the Supplemental Files.

The raw data used for any statistical analyses examining relationships between camera detection and transect survey of dragonflies are available in the Supplemental_data_S1.csv and S2.csv.

The code for statistical analysis using R is available in the Supplemental_code_S1.R

## Supplemental Information

Supplemental information for this article can be found online at http://dx.doi.org/10.7717/peerj.14881#supplemental-information.

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
