# Peer review of "Camera-trapping estimates of the relative population density of Sympetrum dragonflies: application to multihabitat users in agricultural landscapes"

_PeerJ, doi:10.7717/peerj.14881_

## Round 0.1 · original submission · Major Revisions

I agree with all the reviewers' comments.

Reviewer 1 ·

Basic reporting

I appreciate that authors tested their method over three years. However, I think the study has several shortcomings that must be addressed before it can potentially be accepted. First, I believe the conclusions are overstated. Second, I have some doubts about the methodology used to monitor exuvial abundance. Third, the discussion is very superficial, not really discussing advantages and shortcomings of the camera trapping compared with standard monitoring methods, especially with regard to work intensity and time requirement, and considering you may reliably identify only one of many Sympetrum species. I was also wondering what can be monitored by camera traps (that is, what is the applicability of the method) if you can reliably distinguish only one species. Details are given below.

Experimental design

155 – Why did you use three types of camera traps? Could you explain it briefly in the methods? I also miss some discussion about the effect of camera type on its performance (otherwise, if there were no differences, there would be no point in using three camera types).

194–196 – what is the advantage of camera traps over e.g., four-times-per-year survey counting relative abundances of adults at the site (standard monitoring schemes, e.g., https://doi.org/10.1111/icad.12491), especially when considering that the trapping system is not automatic and visual inspection of each image is necessary for correct assessment? Could you estimate in some standardized manner how many e.g., person-hours are needed for each type of survey (that is, camera trapping – including camera maintenance, visual inspection of the images, etc. vs physical inspection of the site counting relative abundances – four times per season, as recommended in standard monitoring schemes). Discussion on this topic is necessary!

276 – exuviae were sampled in early summer, adults (transect surveys) in late autumn. What is life history of S. infuscatum and other darter species found in rice fields? In genus Sympetrum there are species with synchronized hatching (most individuals in the population emerge within a few weeks) as well as those with hatching spread over the whole season. Could your transect survey (exuvia sampling) capture all species with the same probability? And could possible interspecific differences explain your lack of correlation in photographic survey and exuvial surveys? If not further research, then at least discussion on this topic is more than needed. Also, what about male territorial behavior? Do study species exhibit territoriality? If so, photographic survey could be biased by the same individuals guarding their territory around the photographic trap.

Validity of the findings

Lines 37–40 The statement “significant correlation was observed between the camera-detection frequency of mature adults and exuviae-density index in Sympetrum infuscatum; however, a similar correlation was not observed for other darter species.” is valid only for S. infuscatum and exuviae detected in the following year, right? Since in the results you state that “A GLMM revealed significant positive relationships between the camera-detection frequency and adult-density index by line transects (p = 0.000209 and 0.00296, respectively) for both S. infuscatum and other darter dragonflies.“ Therefore, there was a significant correlation for adult density in all study species, and the statement in lines 40-41 is incorrect.

Lines 40–41 “These results suggest that terrestrial camera trapping is effective for monitoring the density of multihabitat users with low mobility.” I would disagree. I don’t expect for example Zygoptera to use those “camera perches”. So low mobility is not a good unifying character. This comment is related to the lack of discussion about advantages, disadvantages, and applicability of your method (see below).

327–328 Well, it is possible to distinguish S. infuscatum from the pictures, but other Sympetrum species are treated together in one “mixed” sample. What are the advantages of having several species mixed-up together in monitoring programs? Or what is the informative value and usability of this mixed sample? Again, the discussion is more than needed.

394–96 I wouldn’t make any conclusions about population dynamics as you didn’t investigate it in your study (e.g., by mark-recapture). Also, I believe that some mark-recapture study is necessary to make more meaningful conclusions about usability of camera traps in monitoring population dynamics (at least discuss the possible methods of verification of the camera trapping performance).
405 – what is the possibility/future of camera trapping regarding potential species-level identification? I miss some solid discussion on this topic.

As a general comment needed to be addressed in the discussion/conclusions: What is the usability of camera trapping method? I see the potential benefit in using it in homogeneous habitats such as rice fields which are dominated by darters, but what about other habitat types and other percher species?

Additional comments

Introduction – certain pieces of information are repeated – namely in lines 53-55 and 64-66, and 72-75 and 92–95. I know you present a particular problem from a slightly different perspective (camera trapping and biodiversity conservation efforts vs camera trapping and multihabitat users in agricultural landscapes; camera trapping and multihabitat users in aquatic environment vs camera trapping and population dynamics of mutihabitat users in agricultural landscapes), but it is still basically the same. I suggest mentioning each piece of information only once and in a comprehensive manner (combine them together).

101 – mature adult darter dragonflies at the terrestrial life stage in an agricultural landscape – mature adult dragonflies are inherently a terrestrial life stage, so the latter part of this information is redundant (and we also know that it is in agricultural landscape – it is in the introductory part of the aims)

116 – I believe that link to Japan Meteorological Agency should be in references rather than in the main text; the same applies to line 120

131 – Did you detect only males in your photographs? You do not mention sex-related differences, I can only see that you sampled females during your transect survey (line 250). Did you expect females used those perches, too?

132 The connection between the sentences is weird. At least, I would change the ending of the first sentence to “based on the following principle.”

275–276 “The investigators visited each plot at least twice during the daytime, from late June to early July in 2018-2020“ So each plot was visited twice a year? Or in total? Well, now I see it from lines 280-281, but from your previous sentence it is very unclear, please clarify.

332 You mention autumn darter here for the first time, but you do not mention that it is S. frequens (well, you mention the scientific name in line 225 but most readers (including me) won’t remember the species identity hidden behind the common name “autumn darter”)

334–335 exuviae of the species detectable on the pictures (S. infuscatum) were scarce (detected only at two plots). But adults were detected in multiple plots, right? Why? Discuss.

359 “adult” should not be in italics

·

Basic reporting

The study by Yoshioka et al shows that custom camera traps can be used to detect the presence of Sympetrum dragonflies in rice fields. The paper is well written, with a clear structure and supported by a clearly stated hypothesis. The gap identified is clearly stated, and the results promising.

Experimental design

The research is original, and based on the previous results from the same team, previously published in PeerJ. It clearly falls into the scope of the journal.
The main idea that the authors want to test is the feasibility to use camera trapping to detect the density of Sympetrum dragonflies, which could be used as an index of habitat health in agricultural environments (rice fields). The methods describe in detail the spatial disposition of the camera traps and the limitations encountered when used for long periods of time.

Validity of the findings

The findings of this study are promising and support the use of this technology for monitoring of agricultural biodiversity in rice fields. The method is particularly suitable to perchers like Sympetrum, but may not be used for other dragonflies, which tend to perch in other kind of substrates, or for other insects which do not show this site attachment to water bodies. This might be discussed.
One limitation of this method is the possibility to lose cameras due to vandalism. Certainly, the camera traps used in this study are clearly visible for any one approaching the paddy fields. Apparently, no camera was lost due to this, but I am worried that this methodology cannot be applied in other places of the world. Did the authors inform not only the owners of the plots but also their neighbours about the study and the value of the methodology for environmental monitoring? Do the authors think that camera losing is a limitation?
Furthermore, no information is given about the time needed to check the cameras and the pictures taken. Over 23,000 pictures with a darter in it is certainly a large number, which needs some time to process. How this effort compares to the effort needed to complete the transect surveys for the adults?

Additional comments

Over the text, I recommed to use the scientific names of the species, to avoid ambiguity. "Autunm darter" and similar names are confusing for readers not familiar with the common names.

On table 1 check the spelling of infuscatum.

I want to congratulate the authors for their innovative techniques applied to biodiversity monitoring. The paper is a good piece of research.

---

## Round 0.2 · accepted · Accept

The paper can be accepted.

Reviewer 1 ·

Basic reporting

Dear authors, I much appreciate your careful and thorough approach to revision, reflecting my and the other reviewer's comments and suggestions; thank you very much for it. I believe the manuscript has benefited from your edits. I really like your study now, and I hope it will become an important baseline for further improvement of the camera trapping method. Well done!

Experimental design

no comment

Validity of the findings

no comment

Additional comments

no comment